# Tissue-Level Flammability Testing: A Review of Existing Methods and a Comparison of a Novel Hot Plate Design to an Epiradiator Design

**Joe V. Celebrezze** [1,*] , **Indra Boving** [1,2] and **Max A. Moritz** [3,4]

1 Earth Research Institute, University of California, Santa Barbara, CA 91336, USA
2 Department of Ecology, Evolution, and Marine Biology, University of California, Santa Barbara, CA 91336, USA
3 University of California Cooperative Extension, Oakland, CA 94607, USA
4 Bren School of Environmental Science and Management, University of California, Santa Barbara, CA 91336, USA
* Correspondence: celebrezze@ucsb.edu

**Abstract:** Increased wildfire frequency and size has led to a surge in flammability research, most of which investigates landscape-level patterns and wildfire dynamics. There has been a recent shift towards organism-scale mechanisms that may drive these patterns, as more studies focus on flammability of plants themselves. Here, we examine methods developed to study tissue-level flammability, comparing a novel hot-plate-based method to existing methods identified in a literature review. Based on a survey of the literature, we find that the hot plate method has advantages over alternatives when looking at the specific niche of small-to-intermediate live fuel samples—a size range not addressed in most studies. In addition, we directly compare the hot plate method to the commonly used epiradiator design by simultaneously conducting flammability tests along a moisture gradient, established with a laboratory benchtop drydown. Our design comparison addresses two basic issues: (1) the relationship between hydration and flammability and (2) relationships between flammability metrics. We conclude that the hot plate method compares well to the epiradiator method, while allowing for testing of bigger samples.

**Keywords:** pyro-ecophysiology; leaf flammability; shoot flammability; flammability methodologies; epiradiator; hot plate; live fuel moisture; water potential





## 1. Introduction

Determining mechanistic explanations for wildfire dynamics is increasingly important as wildfires continue to occur more frequently over larger areas [1,2], jeopardizing more people living in the wildland-urban-interface [3]. The field of pyro-ecophysiology seeks such mechanistic explanations, shifting the focus of wildfire research from the landscape-scale towards cellular and plant-scale processes, linking fire ecology with plant physiology [4]. Many pyro-ecophysiology studies investigate tissue-level flammability, ranging in scale from portions of leaves to large segments of plants. Different methods exist for studying samples along this range. For studies that burn smaller samples (e.g., individual leaves), such as those using epiradiator-based methods (e.g., [5,6]; see Figure 1a,b), flat-flame burners ([7]), or oxygen bomb calorimetry (e.g., [8,9]), differences in flammability may be easier to link to specific mechanisms due to the controllable environment and limited variation between samples. However, challenges arise when applying these conclusions to broader scales due to the lack of realism regarding fuel morphology, as critics of laboratory flammability testing have noted [10]. Other methods accommodate large (~70 cm long) segments of plants, such as the 'grill' method ([11]; see Figure 2b(i)), while some burn entire plants (e.g., large-scale wind tunnels; [12]). These methods more closely emulate

natural fuel structure [10] relative to smaller-scale methods. However, it can be challenging to control environmental factors in larger chambers, and they can be costly to obtain or difficult to build (e.g., large-scale wind tunnels [12,13]). To balance feasibility, controllability, and realism, there is a need for intermediate-sized methods that can burn samples ranging from a single leaf to a small branch sample.

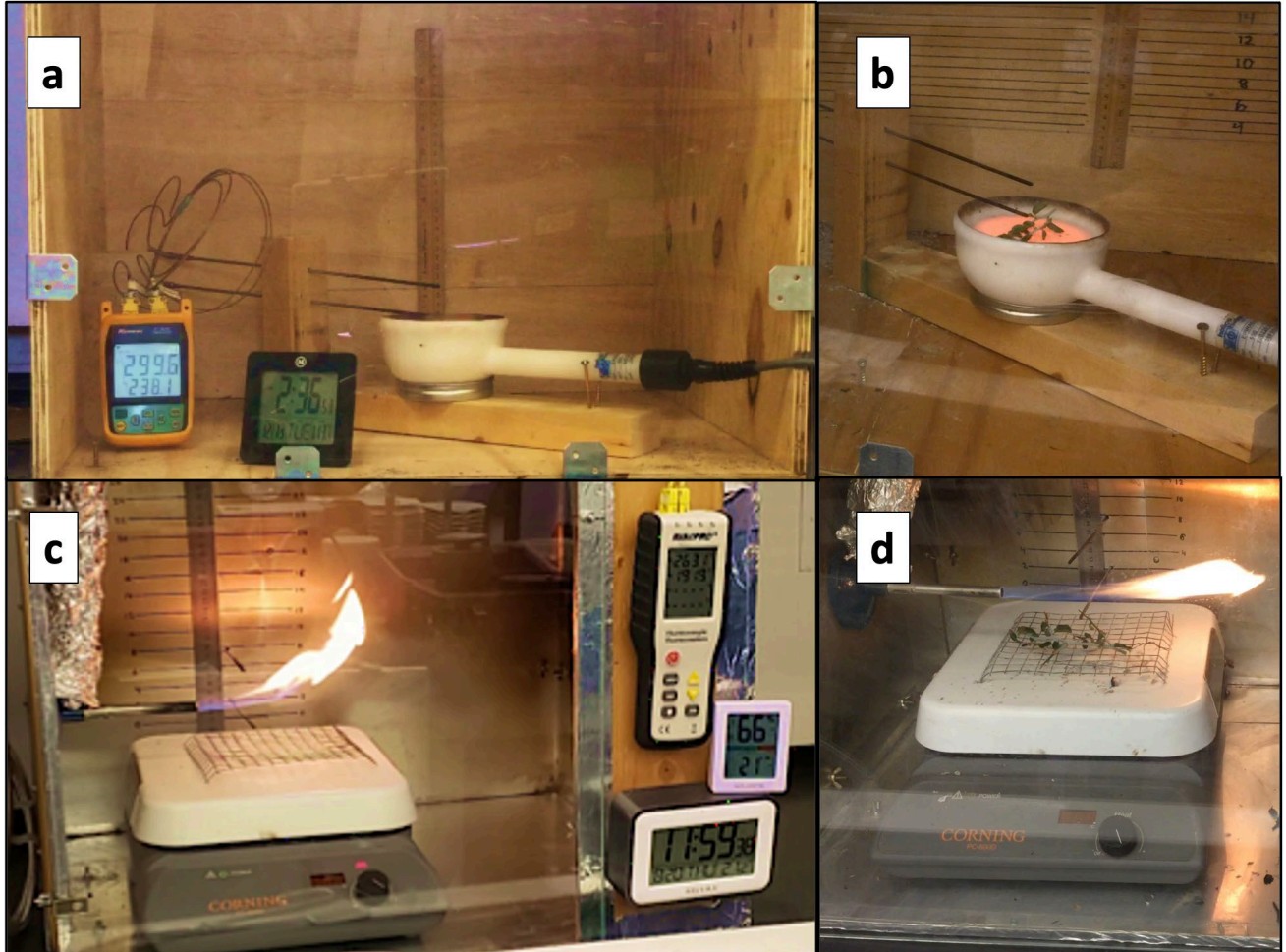

**Figure 1.** Images of the (**a**,**b**) epiradiator and (**c**,**d**) novel hot-plate-based flammability chamber designs used in this study.

Flammability tests often examine the interaction of plant hydration and different flammability characteristics [7,14,15]. Some studies focus on secondary effects of hydration on flammability, typically chemical or physical properties of leaves or plants, including volatile organic compounds [16–18] fuel bed bulk density [19,20], and cuticular wax content [6]. These primary and secondary effects play important roles in flammability, informing management decisions and improving our understanding of interspecific differences [21].

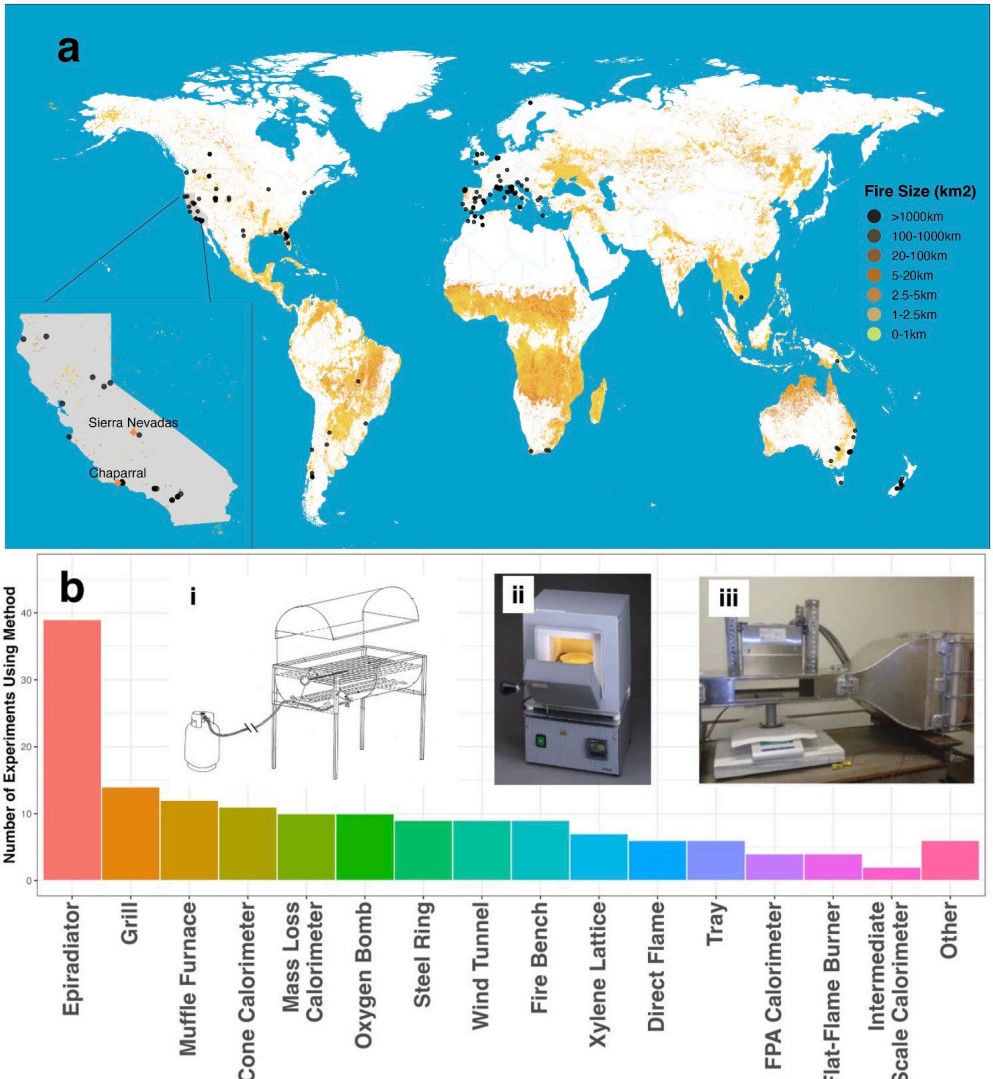

**Figure 2.** (**a**) Map showing the sampling locations of studies included in the literature review (black points), global ignition data from 2015 and 2016 (Global Fire Data), and the corresponding fire size for each fire (shown in a gradient from orange to red). Points vary in their geographical accuracy, as some studies present specific coordinates, while others present broad regions. A total of 13/134 studies did not provide a location specific enough to include on this map. Included also, is an inset map of California showing sampling sites as identified by the literature review as well as our sampling sites (shown with orange diamonds) (**b**) The number of experiments using each of 16 methods across 134 studies investigated by the literature review. Shown also, are images of 3 commonly used methods ((**i**): grill [11]; (**ii**): muffle furnace [22]; (**iii**): wind tunnel [23]). For descriptions of each method, see Table A1 or [24].

Our goals are to characterize methods used to study flammability, compare identified methods to a newly developed hot-plate method (Figure 1c,d), and test the efficacy of the introduced method. We characterized existing methods with a literature review of published studies on tissue flammability and identified needs to inform our novel design. With the literature review, we also consider the reasoning behind the lack of standardization between methods [4,25,26], the geographic range of tissue-flammability studies and how it compares to global ignition data and compare existing methods focusing on small-to-intermediate scale tissue-flammability to the hot plate method. To test the efficacy of this new device, we compared it to an epiradiator design by conducting simultaneous flammability tests using both methods with an emphasis on primary effects of hydration

(i.e., live fuel moisture and water potential). Although multiple methods have been used in the same study on other occasions [12,17], side-by-side comparisons of method performance are largely missing from the literature. With the side-by-side comparison, we specifically addressed the following questions: (1) How do the hot plate and epiradiator methods compare when analyzing the relationship between plant hydration and flammability? (2) How do methods compare when looking at the relationships in a suite of commonly utilized flammability metrics?

## 2. Materials and Methods

### 2.1. Literature Review

To identify, understand, and evaluate existing methods of testing tissue- or plant-scale flammability, we compiled an extensive list of studies in a literature review. We used combinations of the following keywords: "flammability", "methods", and "plant tissue", as well as specific flammability designs (e.g., "epiradiator" and "cone calorimeter"; see Table A1 for other examples) for the initial search, and then used citations from those papers to find other sources. This methodology led to a comprehensive list of 134 tissue- or plant-level flammability studies across a wide range of locations, looking at many different species and functional groups. From these studies, we obtained both quantitative and qualitative data. Quantitative data included sampling location(s) (Figure 2a), the size (either mass, length, area, or volume) of the sample(s) being tested (Figure S1), and heating information (temperature or irradiance). Qualitative data included details on the design of the flammability method(s) used, research goals, justification given for the use of the method(s) over alternatives, and details on the drydown methods (if necessary). The flammability method data was used to sort the experimental methods into 16 groups, defined primarily by the heat source and surrounding structure/chamber (see Table A1 for further details). On six occasions, methods were utilized that were not seen in any other study, so these methods were aggregated into an 'other' group. There were 19 studies that utilized more than one method to test different aspects of flammability or different fuels, with a few using more than two methods [9,12,27].

### 2.2. Laboratory Flammability Tests

#### 2.2.1. Species-of-Interest and Sampling Sites

To compare the hot plate and epiradiator methods, we conducted flammability tests on the live fuel of four species across two regions that are marked by frequent or semi-frequent and intense wildfire: southern coastal California chaparral and mid-elevation mixed conifer forests in the Sierra Nevadas. Live fuel samples allowed us to delve into potential physiological mechanisms driving landscape-level changes in wildfire. Within each region, we studied two shrubs that represented major fuel components of the local vegetation: in the chaparral, *Adenostoma fasciculatum* (chamise) and *Ceanothus megacarpus* (bigpod Ceanothus); and in the Sierra Nevada mountains, *Arctostaphylos patula* (greenleaf manzanita) and *Ceanothus cordulatus* (whitethorn Ceanothus). We gathered live fuel vegetation samples from two sites in the chaparral, north of Santa Barbara, CA in the foothills of the Santa Ynez mountains (Site 1: 34.4613 N, 119.6934 W, elevation: 354 m; Site 2: 34.4931 N, 119.7910 W, elevation: 565 m). Soils at these sites are primarily the Maymen series of Typic Dystroxerepts and the Lodo series of Lithic Haploxerolls, with significant rock outcropping [28]. The Sierra Nevada sampling location in Kings Canyon National Park consisted of two sites (Site 1: 36.7209 N, 118.9708 W, elevation: 1929 m; Site 2: 36.7153 N, 118.9682 W, elevation: 1858 m). Soils at these sites are primarily the Tharpslog series of Vitrandic Humixerepts with occasional rock outcropping [28].

#### 2.2.2. Sample Collection

Samples were collected from the chaparral sites in September 2020 and the Sierra Nevada sites in October 2020—in both cases, sampling coincided with periods of increased fire danger and drought stress. Sample collection consisted of cutting samples of variable

lengths (based on their average xylem length) shortly after sunrise and placing them in two layers of plastic bags in a moist and cool environment to limit moisture loss until sample processing in the laboratory began approximately 24 h later. Sampling took place shortly after sunrise to avoid harsh sunlight introducing xylem cavitation and to ensure the plants were as close to full hydration as possible, limiting interference during the later rehydration process.

### 2.2.3. Plant Hydration Measurements and Laboratory Benchtop Drydown

In the laboratory, samples were cut underwater—avoiding xylem cavitation—to shorter lengths. At least four branchlets were present in each sample for the following: a water potential measurement, a live fuel moisture (LFM) measurement, and for burning in both the epiradiator and the hot plate chambers. Samples were then rehydrated by placing the cut end of the shoot into water in a cool, dark environment for at least two hours until the water potential was $<-0.2$ MPa, following protocols established for pressure-volume curves [29]. After rehydration, flammability testing began immediately and continued as the samples dried on a laboratory bench (mean ambient humidity of around 58%). Methods similar to this drydown method were used in only six studies identified in the literature review (Table A1); however, we selected it due to its similarities to pressure-volume curve protocols and its establishment of a hydration gradient.

Prior to each flammability test, water potential was measured, and samples were prepared for LFM measurements. LFM was measured by weighing the fresh weight of a sample, taking care to avoid reproductive structures or woody stems following guidelines established by the national LFM database [30,31], loading that sample into tins and placing open tins into a drying oven at 100 °C for 24 h before the dry weight was measured and LFM was calculated using the following formula:

$$\text{LFM} = 100 \times \frac{\text{Fresh Weight} - \text{Dry Weight}}{\text{Dry Weight}}. \tag{1}$$

Water potential was measured using a Scholander Pressure chamber (PMS instruments, Corvallis, OR, USA) following standard protocol [32]. Drydown and testing stopped when signs of necrosis were visible or when samples reached about $-10$ MPa, the limits of the pressure chamber used to measure water potential. Sample weight was measured directly before loading the sample into the flammability chamber.

### 2.2.4. Flammability Testing
Epiradiator Design

This commonly used design (see Figure 2b; Table A1) consisted of an epiradiator heating source (Quartzalliance, 500 W) in a small wooden box with a plexiglass window on the front. This 'chamber' was placed in a fume hood to control the environment between burns. Thermocouples were located at 1 cm and 4 cm above the epiradiator surface to capture temperature data. Various flammability metrics—time to ignition, flame duration, flame height, maximum temperature, glow duration, post-flame glow, time to first glow and pre-ignition glow (referred to as glow to ignition)—were assessed using videos recorded on two iPads (iPad Air 2, MGTY2LL/A) capturing the tests from both a lower angle (to capture flame height) and an upper angle (to capture the rest of the flammability metrics) through the plexiglass window. The chamber held a steady temperature where the bottom thermocouple read around 270 °C at the time that the sample was placed atop the epiradiator. Tests consisted of placing 0.4–0.6 g samples directly atop the surface of the epiradiator and removing them either after all phases of burning were complete or after seven minutes elapsed without ignition (pilot tests revealed that samples are unlikely to ignite after this long of a preheating period). Notably, this design differed from some other epiradiator designs, as it did not involve suspending samples with a wire mesh (as in [16,18]) nor did it involve a pilot flame, which emulates an existing ignition source and ignites flammable compounds being driven from the plant during pre-ignition

(as in [33,34]); however, other studies [35–37] also do not suspend samples or include pilot flames.

Hot Plate Design

The hot plate design used in this study consisted of a hot plate heat source (Corning $10'' \times 10''$ hot plate; Model COR-6795-600D) in a medium-sized wooden box with a tempered glass window on the front. This chamber was placed in a fume hood to control the environment between burns. Thermocouples were placed 1 cm and 7 cm above the mesh surface to record temperature data. The hot plate was set to its maximum temperature of 550 °C leading to a steady temperature of around 270 °C as read by the bottom thermocouple at the time the sample was loaded into the chamber. Flammability parameters were captured using two iPads to video-record tests. Tests consisted of placing 10–12 cm samples atop a brass wire mesh raised 1 cm above the hot plate and removing them after all phases of combustion or after 7 min elapsed.

Notable differences between the hot plate and the epiradiator designs include the following: (1) samples were suspended by a wire mesh when using the hot plate design, leading to more radiative and convective heat exposure than conductive heat, which is in contrast to epiradiator samples that were exposed to greater conductive heating through direct contact of the sample with the epiradiator surface; (2) the hot plate design included a propane-fueled Bunsen burner as a pilot flame located 5 cm above the wire mesh which burned throughout the experiment; and (3) hot plate samples were substantially larger than those burned in the epiradiator design, allowing for increased realism and improving the connection to the landscape-scale.

### 2.3. Statistical Analyses
### 2.3.1. Linear Mixed Effects Models

To investigate the effect of the combustion method on relationships between flammability and plant hydration, linear mixed effects models were built using the lmer package in R (v. 1.4.1717). Live fuel moisture and water potential were included in the model selection as primary predictors, as well as 'dry weight' and 'water weight' metrics. We estimated the dry weight of the sample loaded into the flammability chamber using LFM and the total sample weight by using Equation (1) to solve for dry weight/fresh weight and then multiplying this value by sample weight to calculate the estimated dry weight of the sample (sample weight = dry weight + water weight = fresh weight). Water weight was then calculated by subtracting dry weight from the sample weight. Prior to model selection, multicollinearity was assessed using the variation inflation factor (VIF). In cases where VIF > 5, predictors were considered highly collinear, and the model was not included in the selection [38,39]. Prior to assessing the effects of the primary predictors, potential covariates were investigated. These covariates included species, site, sample weight, sampling date/location, and starting temperature as measured by the bottom thermocouple. When selecting the most parsimonious models using Akaike's information criterion (AIC; [40]), species, site, sampling date/location, and sample weight were selected as covariates; however, in cases with wet weight or dry weight in the model as predictors, sample weight was excluded from models due to high multicollinearity.

For the primary model selection (Table 1), the full dataset was split by method to see between-method differences in the relationships between flammability and primary predictors. In this selection, models always included at least one primary predictor, the selected covariates, and plant ID as a random effect on intercepts, to consider repeated measurements. When presenting results in Table 1, *p*-values were determined using the Kenward-Roger approximation through lmerTest [41,42] and marginal and conditional $R^2$ values were determined by Nakagawa's $R^2$ for mixed effects models [43]. A secondary model selection was conducted using the full dataset with the flammability chamber method as an interaction term with the primary predictor in the models. This selection allowed us to perform analysis of variance (ANOVA) tests to determine if there were

significant inter-method differences in the relationships between flammability and plant hydration and/or dryness. In this case, since the flammability metrics had inherent variation due to the size of the samples (i.e., flame height was almost always significantly higher for the hot plate design than the epiradiator design, as samples were around 10 times larger), all eight metrics were z-scaled and centered according to their methods. Dry weight, wet weight and sample weight were similarly scaled and centered. Both the primary and secondary model selection consisted of selecting the most parsimonious model(s) using AIC scores. Multiple models were considered 'top performing' if ΔAIC < 2 from the model with the lowest AIC score.

**Table 1.** A summary table of the primary linear mixed effects model selection. The top models are presented for each method. Significance is indicated by asterisks (*** $p < 0.001$, ** $p < 0.01$, * $p < 0.05$, as determined by a Kenward–Roger approximation of *p*-value using lmerTest [41]). $R^2$ values are determined using Nakagawa's marginal and conditional $R^2$ for mixed effects models; NA values for the conditional $R^2$ occur when the variance of random effects was equal to zero on some occasions [43]. Predictors are abbreviated as follows: live fuel moisture (LFM), water potential (MPa), dry weight (DW), wet weight (WW), and species (sp.).

| Flammability Metric | Top Model(s) | | AIC | | Marginal $R^2$ / Conditional $R^2$ | |
|---|---|---|---|---|---|---|
| | Epiradiator | Hot Plate | Epi. | Hot Plate | Epiradiator | Hot Plate |
| Time to Ignition | MPa *** + DW * × sp. | MPa + DW ** × sp. | 651.189 | 723.571 | 0.345/0.420 | 0.482/NA |
| Flame Duration | DW × sp. | DW × sp. | 513.814 | 572.756 | 0.126/0.137 | 0.184/0.335 |
| | WW × sp. | WW × sp. | 515.805 | 573.224 | 0.098/NA | 0.192/0.307 |
| | | MPa + DW × sp. | | 573.969 | | 0.192/0.326 |
| Flame Height | MPa + DW × sp. | DW × sp. | 522.753 | 477.555 | 0.187/NA | 0.277/NA |
| | DW × sp. | | 523.215 | | 0.157/NA | |
| Glow Duration | MPa + LFM + DW × sp. | MPa + LFM * + DW ** × sp. | 860.247 | 842.114 | 0.322/0.405 | 0.290/0.343 |
| | DW × sp. | | 860.494 | | 0.325/0.402 | |
| | LFM + DW × sp. | | 862.221 | | 0.333/0.339 | |
| Maximum Temperature | MPa *** + LFM + DW × sp. | MPa *** + DW × sp. | 884.899 | 737.287 | 0.224/NA | 0.404/0.603 |
| | MPa ** + DW × sp. | | 886.737 | | 0.193/NA | |
| Post-Flame Glow | MPa + DW × sp. | MPa + DW * × sp. | 828.631 | 809.005 | 0.343/0.419 | 0.289/0.345 |
| Glow to Ignition | MPa + DW × sp. | MPa + LFM + DW ** × sp. | 653.906 | 737.883 | 0.164/NA | 0.256/0.349 |
| | | LFM + DW ** × sp. | | 739.439 | | 0.254/0.355 |
| Time to First Glow | WW *** × sp. | MPa ** + DW × sp. | 576.627 | 752.668 | 0.419/0.423 | 0.185/0.287 |

### 2.3.2. Removal of Partial Effects Using Remef

Remef [44] is an R package utilized to remove partial effects from linear mixed effects models. To isolate the effects of the primary predictor identified in model selection and the covariates we were interested in for specific visualizations (e.g., method in Figure 3), we utilized remef for visualizations involving regression lines atop a scatterplot (Figures 3 and S2–S4). Removing the effects of covariates and the random effect of plant individual adjusted points along the y-axis, made conclusions regarding the primary predictors and covariates-of-interest more evident in the visualizations. Only continuous covariates or discrete covariates with >2 factors involved were removed using remef.

### 2.3.3. Principal Component Analyses

Both methods used in this study utilize the same eight metrics: time to ignition, flame duration, flame height, maximum temperature, glow duration, post-flame glow, glow to ignition, and time to first glow. Many of these metrics covary; therefore, disentangling the relative contributions of each to flammability can be challenging. Multiple conceptual frameworks have been proposed to combat these challenges [45]; however, the most prevalent is the 4-component model. This framework breaks down flammability into four components: ignitability, combustibility, consumability, and sustainability [46,47]. The eight metrics included in this study involve all four components [45], although we

recognize that some metrics may not represent a single component, and instead represent multiple components or fall between components. We expect that flame duration represents sustainability, time to ignition represents ignitability, and flame height and maximum temperature represent combustibility, while the glow metrics likely represent consumability, but may fall in between components or represent multiple components. Ignitability (aligned with wildfire likelihood) and combustibility (aligned with wildfire intensity) have been identified as chiefly important in evaluations of wildfire risk [48], so studies often focus on these components more than consumability and sustainability. To visualize the relationships between these metrics and scrutinize our expectations of where they fall in the 4-component model, principal component analyses were used.

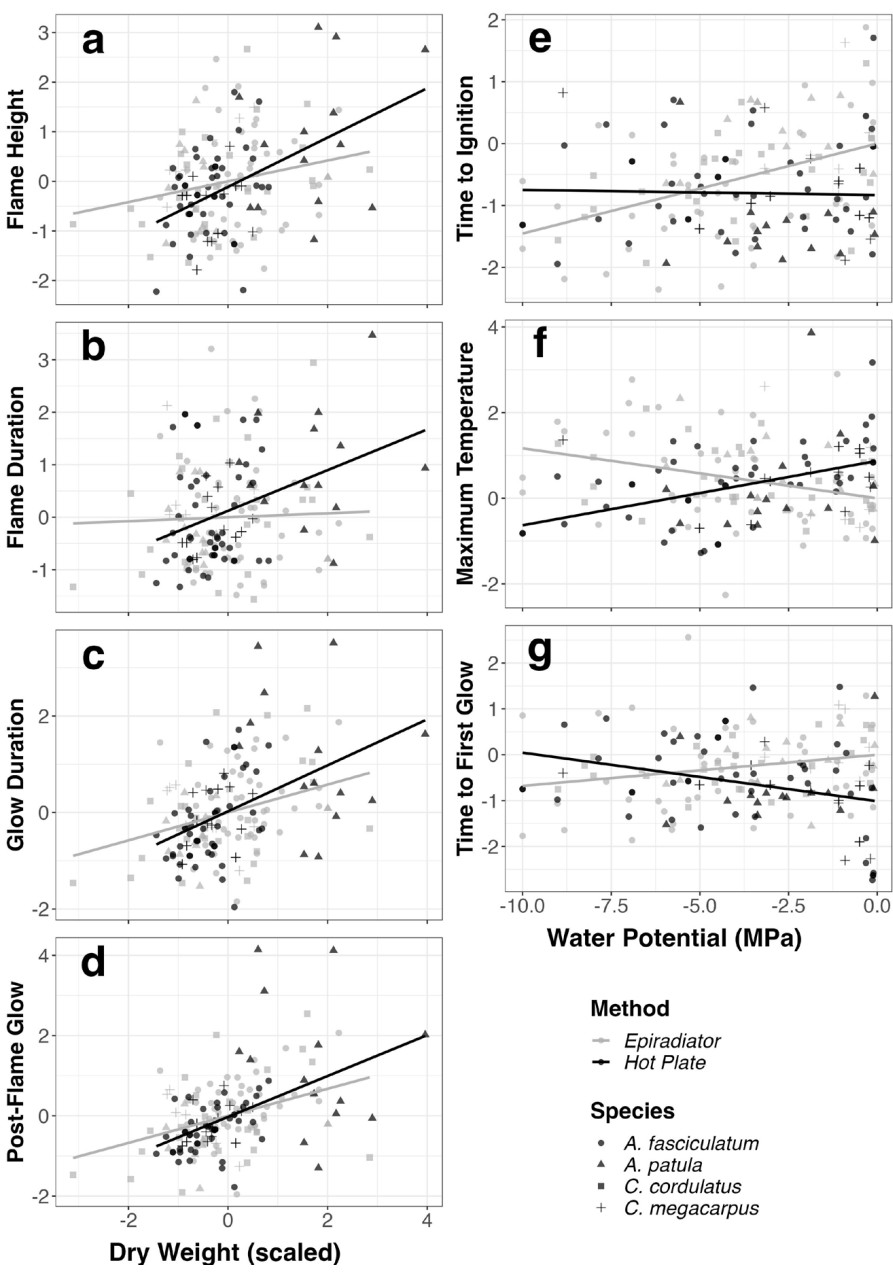

**Figure 3.** Flammability metrics and their best predictors (scaled dry weight (**a**–**d**) or water potential (**e**–**g**)) across species. Remef was used to remove partial effects of covariates, the random effect of individual, and secondary predictors (in certain cases) from the best-performing models to isolate the effects of the primary predictor and the flammability chamber method (epiradiator, gray; and hot plate, black).

Our study compares how the eight metrics fit into the 4-component model for each of the methods. To do this, we conducted two separate principal component analyses (PCAs)—one for epiradiator burns and one for hot plate burns—investigating inter-method differences in the relationships between flammability metrics. In the supplementary index, two other PCAs are presented. These are both for a dataset with epiradiator and hot plate burns, with one of the PCAs conducted on a raw dataset (i.e., without transforming the variables; Figure S6) and the other conducted after flammability metrics are scaled and centered according to the flammability chamber design used (Figure S7). In all cases, PCAs were conducted using the 'prcomp' function from the stats package in R (v. 1.4.1717).

## 3. Results

### 3.1. Literature Review

Through the literature review, we evaluated methods used in 134 studies (Figure 2). Studies ranged in date from 1969 [49] to 2022 [50], with 70% of studies occurring between 2003 and 2019, and with more studies published in 2012 (n = 12) than any other year. Studies varied in location, spanning six continents; however, most studies occurred in Australia, New Zealand, Europe, or the United States of America (Figure 2a). Across these studies, the epiradiator chamber has been used more than any other method (n = 39; Figure 2b). Other common methods for burning vegetation include muffle furnaces (n = 12) and wind tunnels (n = 9), while direct flame (n = 6) and the flat-flame burner (n = 4) were less commonly used. Fire benches (n = 9) and the 'steel ring' method (n = 9), effectively a steel mesh ring in which litter is burned (see Table A1), were the most common methods for burning litter beds, but other methods include the xylene-soaked lattice of strings method (n = 7) and the 'tray' method (n = 6). Many different calorimetry methods—including the cone calorimeter (n = 11), mass loss calorimeter (MLC, n = 10), oxygen bomb calorimeter (n = 10), fire propagation apparatus calorimeter (FPA, n = 4), intermediate scale calorimeter (ISC, n = 2), and a microcalorimeter (n = 1, listed in 'other'; [51])—have been utilized. Calorimetry methods have been used 38 times; however, differences between the methods informed the separation into more specific groups. Lastly, the 'grill' method (n = 14) was developed to burn large portions of plants or full plants [11], and it has been used often since its inception in 2011. Table A1 offers descriptions for each method included in the literature review and points to citations for all 134 studies. There was no indication of hot plates used in flammability chamber designs in any of the studies.

### 3.2. Laboratory Flammability Tests

#### 3.2.1. Flammability Versus Hydration and Dry Weight

Our hot plate and epiradiator testing revealed similar relationships between hydration, dry weight, and flammability (Table 1). For 4/8 flammability metrics (time to ignition, flame duration, glow duration, and post-flame glow), the best linear mixed effects models (the lowest AIC score) had the same combination of predictors for each method. In two of the remaining cases (flame height and maximum temperature), models with identical combinations of predictors exhibited minimal reductions in AIC relative to the selected models ($\Delta$AIC < 2 in both cases). The only flammability metrics without identical predictors in top-performing models ($\Delta$AIC < 2) between the two methods were glow to ignition and time to first glow. Regarding the selection of the predictors, water potential and dry weight were more consistently present in top models than LFM; however, we did end up visualizing the effect of live fuel moisture on the flammability metrics which showed similar trends to water potential for most metrics (Figure S2). Visualizations of all of the flammability metrics' relationships with water potential (Figure S3) and dry weight (Figure S4) are also included in the supplemental index. Due to significant interactions between predictors and species in five of the top models, we also investigated interspecific differences (Figure S5, Discussion S1).

Regarding the secondary model selection, including a 'method' term significantly improved models for time to ignition ($\chi^2$ = 14.656, $p$ = 6.57 $\times$ $10^{-4}$; Figure 3e), maximum

temperature ($\chi^2$ = 23.714, $p$ = 7.09 × 10$^{-6}$; Figure 3f), and time to first glow ($\chi^2$ = 19.692, $p$ = 5.30 × 10$^{-5}$; Figure 3g) indicating significant inter-method differences.

### 3.2.2. Relationships between Flammability Metrics

Principal component analyses (Figure 4) exhibit the relationships between flammability metrics for each flammability chamber. The epiradiator PCA (Figure 4a) and the hot plate PCA (Figure 4b) yielded similar results in terms of the proportion of variation in data explained by each of the first three principal components (Table 2). Other principal components had standard deviations >1, thus we did not consider them in the analysis [52]. In both methods, glow to ignition was negatively loaded on PC1 and flame duration and time to ignition were, respectively, negatively, and positively loaded on PC2 (Table 2). Otherwise, the principal component loadings differed between the two methods. However, the relationships between the flammability metrics as seen on the PCA plots (Figure 4) were relatively consistent across the two methods with differences primarily evident for time to first glow and maximum temperature.

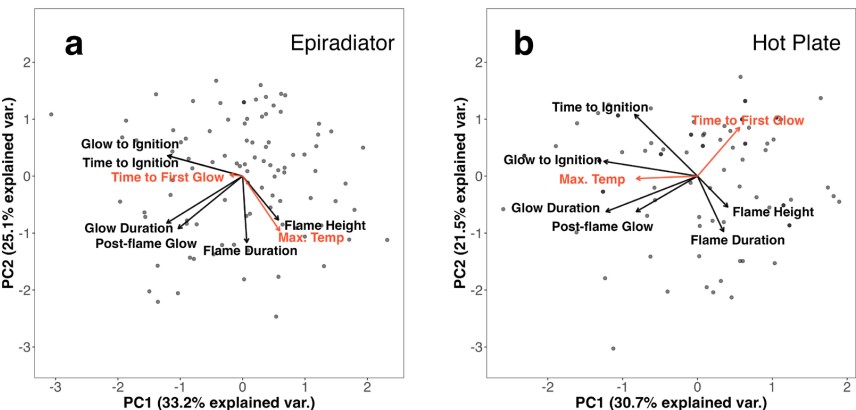

**Figure 4.** Principal component analyses for all measured flammability metrics for both the (**a**) epiradiator and (**b**) hot plate methods. Key differences between the two methods are highlighted with red-orange arrows and text labels.

**Table 2.** PCA loadings following varimax rotation for the two principal component analyses shown in Figure 4. Bolded values represent instances where flammability metrics have loadings with an absolute value of 0.5 or greater indicating a relatively large influence upon that principal component.

|  | Epiradiator | | | Hot Plate | | |
|---|---|---|---|---|---|---|
|  | **PC1** | **PC2** | **PC3** | **PC1** | **PC2** | **PC3** |
| Flame Duration | −0.44 | **−0.62** | 0.14 | 0.02 | **−0.76** | 0.05 |
| Flame Height | 0.03 | **−0.68** | −0.26 | **0.57** | −0.15 | −0.31 |
| Post-flame Glow | **−0.90** | −0.08 | −0.10 | 0.09 | 0.07 | **−0.94** |
| Time to Ignition | −0.44 | **0.59** | **0.52** | −0.41 | **0.81** | −0.08 |
| Max. Temp. | −0.08 | **−0.72** | 0.15 | **−0.64** | −0.06 | −0.12 |
| Glow Duration | **−0.97** | 0.06 | 0 | −0.34 | −0.02 | **−0.89** |
| Glow to Ignition | **−0.51** | **0.70** | 0.11 | **−0.83** | 0.21 | −0.24 |
| Time to First Glow | 0.01 | −0.06 | **−0.96** | **−0.54** | **0.62** | 0.19 |
| Proportion of Variance | 33.19% | 25.09% | 16.04% | 30.72% | 21.54% | 17.94% |
| Cumulative Proportion | 33.19% | 58.28% | 74.31% | 30.72% | 52.26% | 70.19% |

## 4. Discussion

### 4.1. Methods of Testing Tissue-Level Flammability

Through an extensive review of the literature, we determined the most common flammability testing designs (Figure 2b). Epiradiator methods have been used more than

any other method, supporting our comparison of the hot plate method to an epiradiator method and allowing us to understand how the hot plate method could contribute to future pyro-ecophysiology studies.

Although there have been continued calls for the standardization of flammability-testing methods [4,25,26], a comprehensive literature review highlights the importance of specialized flammability chambers designed to meet the needs of specific research questions. While having many different methods to test flammability leads to challenges comparing results and conclusions between studies, different methods have been developed to investigate specific research goals. Studies vary in their interests and scales, ranging from full plants [12,53] to leaf litter layers [20,54] to individual leaves [55,56]. These studies investigate a variety of questions, including but not limited to the effects of slope on wind-driven flames [57], the impacts of a moving flame front [58], how litter bed depth relates to flammability [20], and the relationship between bark thickness and fire resistance [59]. Due to the breadth of research questions being asked in fire science, complete standardization of methods may be unattainable. Furthermore, choosing the incorrect method for a given study could lead to a skewed interpretation of flammability. For example, using a method with a guaranteed ignition source (e.g., xylene lattice of strings, wind tunnel, or tray methods) does not provide any information on ignitability; therefore, any conclusions made using these methods on interspecific differences in flammability would lack this key component. Instead, studies similar in scale and objective might seek method standardization in their niche of the field. As a central example, we believe that the 'grill' method [11] is rapidly becoming the standard method to burn larger samples due to its large capacity and—therefore—ability to burn plants that more closely emulate natural fuel structure, flexibility to address a range of questions, and the relative ease in which it can be built.

While investigating flammability using small to intermediate-sized live fuel samples, such as those best suited to our hot plate method, previous studies have typically utilized the following methods: epiradiator, muffle furnace, calorimetry, small-scale wind tunnel, and flat-flame burner. Since wind tunnel and calorimetry designs explicitly measure wind and heat release (Table A1), they have slightly divergent research goals than the hot plate, epiradiator, muffle furnace and flat-flame burner designs; therefore, the following discussion compares the hot plate design to the other three methods used for measuring tissue-flammability (epiradiator, muffle furnace, and flat-flame burner).

Looking specifically at these three methods, the temperature of the heating source varied widely (468 ± 166 °C), and the hot plate method had a comparable *surface* temperature to these methods (about 550 °C). An overarching advantage of the hot plate design compared to the epiradiator, muffle furnace, and flat-flame burner designs is the ability to burn samples ranging from a single leaf to a small branch. For the majority (65%) of epiradiator studies, sample weights were ≤1 g (Figure S1) and for the muffle furnace and flat-flame burner designs, samples were most often 'individual leaf specimens'. In this study, the hot plate design burned samples ranging from 0.72 g to 6.10 g, with lengths ranging from 10–12 cm; however, it is capable of consistently burning samples greater than 20 g. Furthermore, the hot plate design can more feasibly scale up relative to the epiradiator and muffle furnace designs. Multiple hot plates could easily be placed in a larger chamber with minor adjustments to burn vegetation samples upwards of 100 g. In contrast, epiradiators would be difficult to place side by side due to their round, concave shapes, while the muffle furnace method involves placing a sample inside the muffle furnace, which is limited in size. The hot plate's ability to accommodate larger samples, thereby further incorporating morphology and plant structure into flammability tests, makes it better suited to test live vegetation representative of plants found on the landscape.

Some more specific advantages of the hot plate method exist when comparing it to the muffle furnace and flat-flame burner. First, muffle furnace tests take around 1–2 h to preheat which contrasts with the 20–30 min of preheating necessary for the hot plate design. In addition, muffle furnace tests must be run with an open door to record flammability

metrics, leading to potentially uneven airflow and fluctuations in temperature throughout testing. Likewise, the flat-flame burner designs do not have surrounding chambers, leading to immeasurable variation between tests and introducing challenges replicating the exact study conditions elsewhere. The main advantage of using a flat-flame burner design is the representation of a moving flame front which the hot plate design does not incorporate, so studies looking explicitly at the dynamics of a moving flame front may prefer the flat-flame burner design over alternatives such as the hot plate design.

While we centered our focus on advantages of the hot plate method over alternatives, the literature review also revealed a discrepancy between tissue flammability study locations and global fire occurrence (Figure 2a). Specifically, Sub-Saharan Africa, Southeast Asia, and South America have high frequencies of ignitions, yet very few studies evaluate tissue flammability in these regions. This discrepancy is likely due to the lack of funding sources and publishing opportunities in certain regions of the world, as well as differences in social and economic costs associated with wildfire. Some regions of the world may have less funding and publication opportunities than high income regions such as North America, Europe, and Australia [60], where we see most tissue flammability studies in our literature review. Although this certainly relates to economic and social inequalities, in the specific case of fire ecology studies, it could also relate to the social and economic costs associated with wildfires in these regions [61]. Although Africa has the majority of ignitions, less people are affected by wildfires and the economic cost of wildfire is lower than any other continent (except for Antarctica) [61]. Differences in fire severity or intensity [62] could also explain differences in where ignitions occur versus where studies take place as well as differences in social and economic costs associated with wildfire [60]. Regardless, if we hope to understand global dynamics of wildfire, we—as a field—should advocate for increases in funding for labs in underprivileged regions, so that more researchers may continue to investigate questions like the ones discussed here.

### 4.2. How Do the Methods Compare When Analyzing the Relationships between Plant Hydration, Sample Dry Weight, and Flammability?

To compare the hot plate and epiradiator methods, we juxtaposed relationships between moisture and flammability metrics for each of the two methods. Although we aimed to investigate the effect of plant hydration on flammability, we determined that sample dry weight was also an important predictor of flammability metrics for both methods (Table 1). Additionally, the two methods had identical combinations of predictors for the majority (6/8) of flammability metrics.

Looking at the trends in the relationships between flammability and hydration, we determined that significant inter-method differences existed for time to ignition, maximum temperature, and time to first glow. For time to ignition, in the epiradiator design, drier samples tended to ignite more quickly, while in the hot plate design there was no significant change in ignitability as the samples dried (Figure 3e). For maximum temperature, drier samples burned hotter in the epiradiator design, while for the hot plate design, wetter samples burned hotter (Figure 3f). In both cases, differences between the two methods may partially be explained by each method's ability to capture the influence of volatile organic compounds (VOCs). VOCs are often stored and/or emitted by plants and influence the flammability of leaf litter [17] and live vegetation [37,51] due to their low flash point, high heating value, and low flammability limit (LML). However, VOCs are also created by the volatilization of organic compounds in any fuel source, regardless of VOC storage or emissions, during the pre-ignition phase.

For this reason, we expect the hot plate design, which includes a pilot flame, to incorporate the impact of VOCs more than the epiradiator design. Studies on essential-oil-bearing herbs [63,64], cut grass [65], wood building products [66], and pine foliage and litter [48] show variable VOC compositions and content in plant tissue as samples dry. The finding that wetter samples burned hotter in the hot plate design was unexpected; however, we suspect that the variability of VOC content over the course of the benchtop drydown

offers a plausible explanation. During the benchtop drydown, we believe VOCs are emitted but not replenished leading to a tradeoff between dryness and VOC concentration; however, this should be backed up with further evidence looking specifically at species such as ours that are not known to store large amounts of VOCs, but instead may emit VOCs as they are being synthesized. Regardless, the hot plate design includes a pilot flame that ignites VOCs and other flammable compounds actively emitted by the sample, so the impacts of VOCs might be enhanced, while—in the epiradiator design—ignition is largely driven by the dryness of the sample. In the hot plate method, the concentration of VOCs relative to water vapor driven from the sample must reach some threshold for the pilot flame to ignite the sample and, if the VOC content is higher (in wetter samples), we expect hotter flames due to the high heating value of VOCs. This might explain the differences in maximum temperature trends between methods.

We suspect that the two methods' different heating regimens may result in the differences observed for time to first glow. As the hot plate method employs a wire mesh to suspend samples, it heats samples with convective and radiative heat more than conductive heat. The epiradiator method has direct contact between the sample and the surface of the epiradiator; therefore, it depends more on conductive heat. For hot plate burns, 31/83 ignitions had no glow time prior to ignition, contrasting with the epiradiator design which always glowed prior to ignition. We expect that the points of contact between the sample and the epiradiator surface improves the likelihood of smoldering, potentially explaining differences in time to first glow. This could also explain the inter-method differences in glow to ignition identified in the primary mixed effects model selection.

### 4.3. How Do the Methods Compare When Looking at the Relationships in a Suite of Flammability Metrics?

For the epiradiator design (Figure 4a), the flammability metrics align closely with the 4-component model [46,47], where ignitability is represented by time to ignition and glow to ignition (loaded on both PC1 and PC2; Table 2), combustibility is represented by flame height and maximum temperature (loaded on PC2), sustainability is represented by flame duration (closely related to combustibility, loaded on PC2), and consumability is represented by post flame glow and glow duration (loaded on PC1). Time to first glow is heavily loaded on PC3 rather than PC1 or PC2, thus it is difficult to place into any of the four axes, although we would expect it to relate to other glow-metrics. The hot plate design has conclusions that slightly deviate from the 4-component model (Figure 4b). Flame duration, flame height, time to ignition, glow to ignition, post-flame glow and glow duration all show similar relationships relative to the epiradiator PCA. However, maximum temperature no longer clusters with flame height as a combustibility metric and time to first glow is opposite to other glow metrics. This reflects inter-method differences for the relationship between flammability and plant hydration. In the hot plate PCA, time to first glow and time to ignition are loaded similarly on PC2 (0.62 and 0.81, respectively; Table 2). This could result from the 31/83 ignitions in which time to first glow and time to ignition are equivalent. Maximum temperature is also more closely related to the ignitability metrics in the hot plate PCA, further supporting their shared relationship with VOCs. This relationship is quantitatively supported by the loadings of maximum temperature, time to ignition and glow to ignition on PC1 ($-0.64$, $-0.41$, and $-0.83$, respectively). This has a stark contrast with the loadings of maximum temperature and time to ignition on PC2 for the epiradiator method which we expect VOCs to have less of an impact (maximum temperature: $-0.72$ and time to ignition: $0.59$).

### 4.4. How Does the Hot Plate Method Compare to the Epiradiator Method?

In many cases, the two methods compare well. Usually, identified differences could be explained by the inclusion of a pilot flame in the hot plate design or the differences in heating regimens between the two designs. For a more comprehensive study comparing two methods such as these, further measurements on secondary effects should be consid-

ered, such as VOC concentration and composition. Likewise, isolating differences between two methods rather than having multiple differences could help explain what exactly is driving divergent conclusions between methods (in our case, the two designs had three distinct differences). Regardless, the comparison between these two methods provides evidence that the hot plate method should be used in further studies and will produce conclusions comparable to previous ones, while better representing vegetation flammability on the landscape and further bridging the gap between leaf-scale and plant-scale flammability testing.

## 5. Conclusions

As the field of pyro-ecophysiology seeks to connect tissue-scale studies to the landscape-scale, burning larger vegetation samples under realistic heating scenarios is of utmost importance. Relative to alternative small-to-intermediate methods, the hot plate method has advantages of size, cost, attainability, and replicability. Through a comparison with the most used method, the epiradiator, we conclude that the two methods are similar in more ways than they are not. Furthermore, their apparent differences have plausible explanations grounded in differences between the two chamber designs—e.g., the pilot flame and the form of heating. We find this novel method to be most useful in research investigating the flammability of live vegetation samples, and we hope it fills an important gap in the study of potential mechanisms driving wildfire dynamics.

**Supplementary Materials:** The following supporting information can be downloaded at https://www.mdpi.com/article/10.3390/fire6040149/s1: Figure S1: Literature Review—summary of sample weights for flammability tests; Figure S2: Live fuel moisture vs. raw flammability metrics; Figure S3: Water potential vs. raw flammability metrics; Figure S4: Dry weight vs. raw flammability metrics; Figure S5: Interspecific differences between *A. fasciculatum* and *A. patula*; Discussion S1: Could interspecific differences help explain inter-method differences? Figure S6: PCA—raw flammability metrics; Figure S7: PCA—flammability metrics scaled and centered by method.

**Author Contributions:** Conceptualization, J.V.C., I.B., and M.A.M.; Methodology, J.V.C. and I.B.; Data Curation, J.V.C. and I.B.; Formal Analysis, J.V.C. and I.B.; Funding Acquisition, M.A.M.; Software, J.V.C. and I.B.; Visualization, J.V.C. and I.B.; Writing—original draft, J.V.C.; Writing—review and editing, J.V.C., I.B., and M.A.M. All authors have read and agreed to the published version of the manuscript.

**Funding:** This research was funded by the University of California's National Laboratories (UCNL) Laboratory Fees grant program under grant number LFR-18-542511 (as a part of the California Ecosystems Futures project).

**Institutional Review Board Statement:** Not applicable.

**Informed Consent Statement:** Not applicable.

**Data Availability Statement:** The data presented in this study, along with the code necessary for statistical analyses, data visualization, tables, and the metadata are openly available at https://github.com/celebrezze/flam-methods-comparison (accessed on 15 February 2023). In the metadata and README associated with the attached GitHub repository, two external publicly available datasets are noted: for global ignition data—https://daac.ornl.gov/cgi-bin/dsviewer.pl?ds_id=1642 (most recently accessed on 22 March 2023) and for precipitation data from Santa Barbara County—https://www.countyofsb.org/2256/Historical-Rainfall-Reservoir-Informatio (accessed on 26 February 2021).

**Acknowledgments:** We would like to acknowledge Kristina Fauss and Isaac Park for their contributions in the editing process, Leander Love-Anderegg for providing counsel regarding exploratory data analyses, and Aaron Ramirez and Ryan Salladay for initially developing the laboratory protocol regarding flammability tests using the epiradiator method and the laboratory drydown method. We thank the anonymous referees for their input on the manuscript.

**Conflicts of Interest:** The authors declare no conflict of interest.

# Appendix A

**Table A1.** Descriptions of each of the flammability testing methods included in Figure 2b and list of studies using each method; for more comprehensive descriptions of flammability testing methods, see [24].

| Method | Description | Studies |
|---|---|---|
| **Epiradiator** | This method is defined by a Quartzalliance epiradiator as the primary heating source. Suspension methods (i.e., wire mesh), inclusion of a pilot flame, sample weights, heat flux, starting temperature, locations of thermocouples vary across different studies. Studies investigating bark flammability (e.g., [67]) have the surface of the epiradiator in a vertical orientation; however, the epiradiator is typically in a horizontal orientation, as most studies using this method investigate either live vegetation or litter layers. | [5,6,15–18,27,33–37,48,67–91] * |
| **Grill** | The grill apparatus was designed and described in detail in [11] In this case, it involves an 85x60cm metal barrel cut in half, a metal grill placed in between each half with propane-powered burners below the grill as heat source, a blow torch as an ignition source, and removable wind protection in an attempt to control the environment between burns. In some other cases, a manufactured BBQ grill was used in place of the barrel-based grill, as they are more easily accessible. | [11,92–104] |
| **Muffle Furnace** | This method consists of placing a sample inside a muffle furnace and measuring flammability parameters such as time to ignition. Samples are often individual leaf specimens. Temperature of the muffle furnace varied between studies from 400-700°C. The door of the muffle furnace is kept open during tests to allow for the measurement of flammability metrics. | [8,49,55,56,100,105–111] |
| **Cone Calorimeter** | This calorimetry method measures heat release rate by measuring the consumption of oxygen resulting from combustion. It consists of a conical heater with a fixed heat flux to heat samples. The ignition source varies between tests with the more common ones including a spark ignitor and a pilot flame. Tests are typically performed in accordance with ATSM E 1354 standards [112]. | [113] *, [114–121], [51,122] ** |
| **Mass Loss Calorimeter (MLC)** | This is a specific type of calorimeter, manufactured by Fire Testing Technology Limited (FTT®), which is very similar to a cone calorimeter, but with adjustments to specialize in full fires and to measure smaller changes in mass loss. It also looks at heat release and heat flux. The MLC involves a conical heater, a 500g-capacity load cell, a spark ignition source, a chimney with a thermopile. Tests were most commonly conducted at 50 kW/m$^2$. | [9,12,13,50,59,67,123–126] |
| **Oxygen Bomb** | This is a calorimetry method commonly used to measure gross energy content and heat of combustion. It operates by having a reaction occur in a container (referred to as the 'bomb') with a fixed volume, so that changes in temperature can be attributed to the energy flux of the reaction rather to any changes in volume. In this review, it was always paired with other methods. | [6,8,9,12,75,79,89,93,127,128] |
| **Steel Ring** | This is similar to the 'tray' method, but it involves a more sophisticated setup to maximize airflow and can incorporate both leaf litter and/or vegetation samples. It involves a steel mesh ring with samples loaded into it. Often, it also has a fireproof base, such as one made from a hardiflex cement fiber board. Ignition sources vary, including a paper towel soaked in isopropyl alcohol and a cotton ball soaked in ethanol. | [129–137] |
| **Wind Tunnel** | This group consists of any method that has a wind tunnel directing wind towards the samples emulating winds that play a key role during wildfires. Sample size and composition varies across the studies, from looking at full, reconstructed shrubs and litter layers to looking at 4g samples. A heat source, such as an infrared panel, is often utilized and usually located above a sample. An ignition source is typically present and varies by study, with the most common sources including a firebrand and a coiled wire igniter. | [12,13,23,57,138–142] |
| **Fire Bench** | 'This method involves a large flat surface in which large plant samples or, more commonly, litter beds are tested for flammability. Ignition sources vary for the different studies and include 'firebrands' (usually a small block of wood, ignited by means of an epiradiator), cotton wicks soaked in a flammable liquid such as ethanol, and a propane torch. | [17,54,125,143–148] |
| **Xylene Lattice** | 'This is another commonly used method to test litter bed flammability. It involves placing litter beds atop a lattice of cotton strings soaked in xylene, a flammable liquid, and then igniting the lattice of strings from one or multiple points leading to rapid ignition and combustion of the litter bed. | [20,149–154] |
| **Direct Flame** | Samples are directly ignited by an open flame without any other heat source present. The open flame varies across studies from a Bunsen burner to a larger U-shaped gas burner. | [19,127,128,132,155,156] |
| **Tray** | This is yet another method to test litter bed flammability. It involves placing litter beds in a metal tray with limited airflow from the sides or bottom. An ignition source, typically a firebrand, is placed in the center or edge of the litter bed to cause ignition and combustion of the litter bed. | [27,139,157–160] |

**Table A1.** *Cont.*

| Method | Description | Studies |
|---|---|---|
| **FPA Calorimeter** | This method is similar to the cone calorimeter, but the combustion chamber allows for a more controlled environment with less stochasticity in gas flow rate and composition. | [161–164] |
| **Flat-Flame Burner** | This method was designed to emulate a moving flame front. Small samples (typically, individual leaves) are suspended and a flat flame burner moves towards the sample by route of a small motor. Sometimes, a 6000 W radiative panel (Omega) is also included as a radiative heat source. | [7,58,165,166] |
| **Intermediate Scale Calorimeter** | This is also similar to a cone calorimeter, but it is designed for larger scale samples, typically burning full plants. It consists of equipment to measure gaseous concentrations, a propane line burner to create a vertical 'wall of flame', three wooden walls lined with ceramic fibreboard, and a plant rack in the middle of the three wooden walls. | [23,116] |
| **Other** | (1) Litter was placed in a cubic basket which was then loaded into a recirculating air oven which sustained temperatures up to 250 °C, while thermocouples placed in the middle of the litter took temperature readings. (2) This is a specialized experiment investigating the propagation of flames from litter layers to shrub canopies. $20 \times 20$ cm litter samples were dried and then ignited 10 cm below a steel wire netting holding a variety of shrub samples representative of the shrublands of SE France to emulate litter-to-shrub fire spread. (3) Litter was placed in a cubic cage which was loaded atop a sand surface into a 1.09m-tall chamber made of vermiculite insulation board. Inside the chamber, the sample was heated using an IR lamp and a handheld spark generator was used as an ignition source. (4) Leaf-level flammability was measured using a Federal Aviation Administration (FAA, USA) microcalorimeter using 10–15 mg samples. (5) Shoot-level flammability was measured using a horizontal 4 kW radiant panel located 1 cm beneath the shoot sample which was suspended by a specialized holder, with a heat flux sensor measuring heat flux and a scale beneath the sample holder measuring weight consumed during burning. (6) Full-plant flammability was measured using potted plants. Aluminum discs were located at soil level and 50 cm above soil level to estimate heat release. The plant was ignited by a burning cotton ball, soaked in 10 mL of ethanol. | (1) [167] (2) [160] (3) [168] (4) [51] (5) [30] (6) [169] |

\*: the six studies identified with an asterisk used the laboratory benchtop drydown process described in our methods and in pressure-volume curve protocol [29] \*\*: specific to the 'cone calorimeter' classification, in some cases, the iCone calorimeter (Fire Testing Technology, East Grinstead, UK) was used. This is an advanced, automatic cone calorimeter designed to test a variety of flammability parameters.

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
