# Peer review of "Tissue-Level Flammability Testing: A Review of Existing Methods and a Comparison of a Novel Hot Plate Design to an Epiradiator Design"

_fire, doi:10.3390/fire6040149_

Round 1
Reviewer 1 Report
The manuscript provides firstly a literature review on methods developed to study tissue-level plant flammability. Flammability is an important topic in wildfires, therefore the ms. is in the scope of “Fire”. Next, the ms. presents a novel device to analyze tissue-level plant flammability, the “hot-plate-based flammability chamber design”, comparing it to the most commonly used design (as showed by the literature review), the epiradiator design, in order to test the efficacy of the introduced method.
Although several reviews on flammability have already been published, I find the one presented in this manuscript is particularly comprehensive and adequately oriented, and the results clearly presented in the form of a table and figures. Therefore, information provided is a valuable and useful addition to the existing literature in this topic.
Comparison between the novel method proposed and the classical one is well focused, simultaneously conducting flammability tests on live fuel of four species, along a moisture gradient. Statistical analyses are appropriate and Results are properly discussed. According to these results, the proposed laboratory method seems to offer interesting possibilities for the study of flammability at tissue level. Nevertheless, limitations of laboratory tests linked to the scale should be, at least, mentioned (see Ref. [12]).
Besides, there are a few mistakes that need to be corrected and I have some comments to improve the manuscript:
L44: Ref [11] does not include tests on entire plants, but tests on samples of live needles and samples of leaves and fine twigs of understory shrub species.
L322: Figure 3 is too small; it is difficult to appreciate the details
L385-386: this statement is not seen in Figure S3
L417-418: Ref [60] the reference does not seem to be adequate for this statement
L478: the 4-components model instead of 4-axis model
Supplementary Materials:
Figure S1: I do not quite understand what the y axis represents (Perhaps it is because I am not familiar with this type of figures) % of studies?
Therefore, based on my examination of the manuscript, my recommendation is “Minor revisions”
Reviewer 2 Report
Reviewer Report – Fire
Joe Celebrezze et al.
Tissue-level flammability testing: A review of existing methods and a comparison of a novel hot plate design to an epiradiator design
This ms describes a novel method of testing tissue-level flammability (hot plate), and compares it to a commonly used existing one (epiradiator). There is increasing interest in measuring flammability of a range of fuel types, and it is useful to have an additional option. The ms explores how important flammability metrics of four species respond to plant moisture content and sample mass (useful in its own right), and shows there are minor differences between the two methods. The ms is well-written, the data appear to have been carefully collected and the analyses are appropriate. The literature review was useful in setting the scene in a quantitative way. My comments are all quite minor.
Line 43 – where does the ‘flat-flame’ method fit here? It is discussed in some detail in the discussion (see lines 376 &380), but I was unclear what this was. It needs some introduction.
Line 51 – I don’t think ‘however’ is appropriate here – it sets the reader on the wrong track about what to expect in this sentence.
Fig. 1 – excellent to have a photo of the setup. However, I would have appreciated a few photos to illustrate the set up burning some plant samples, and exactly where the plant material sits – perhaps 2 extra panels could be added?
Line 186 – So the plant material was directly exposed to this flame? how long did the pilot flame burn for -the whole experiment?
L223 – I am very pleased to see the sample mass is considered in your study. Far too many studies ignore its potentially large influence. As you acknowledge, much of the difference between epiradiator and hotplate methods could be due to sample size
L240 – Be explicit here about which metric applies to which flammability component. What is your consumability metric? (In practice, consumability appears strongly related to combustibility)
Fig 2 – a minor point, but I find the colours indicating fire size are very difficult to distinguish from each other
L306/ Fig. 3/ Fig. S2. I am a bit puzzled about the relationships you choose to present. You present all 8 flammability metrics in relation to your less important variable LFM (albeit in supporting info), but only 4 in relation to each of water potential and dry weight. Also, you present the non-significant relationship between glow to ignition and MPa, but not dry weight, which was significant for the hotplate method.
You could comment somewhere here that trends are similar whether MPa or LFM is used as the measure of hydration (at least for the 4 metrics shown here for MPa)
Fig. 4 – good presentation using colours to highlight the differences. These patterns are different from some other published ones, presumably because yours incorporate a wide range of live fuel moisture, compared with dry samples only in many other studies. For example, flame height and flame duration are quite closely aligned here, whereas I would expect a negative correlation with dry samples.
Lines 373 – 380 - I would also have liked some discussion about the plant grill – the samples it burns can be bigger again, and possibly more realistic. Your literature review showed this is the second-most common method used.
Lines 376/380 – it would be good to have the flat-flame burner introduced earlier
L388 – are these fresh or dry mass?
L410-422 – Your map doesn’t show fire intensity or severity, which is really important but not necessarily reflected in area burnt. For example, vast areas of savanna are burnt globally each year, often intentionally. Much more concerning are the dangerous, intense but less frequent wildfires in temperate forests.
L434-436 vs 437-438 – These two sentences are apparently contradictory – you need to help the reader a bit here.
L440-441 – wetter samples burning hotter is completely unexpected. You do offer a possible explanation in discussing the differences between the two methods, but you also need to explicitly address this unexpected finding in its own right – even if just by inserting a sentence along these lines.
Fig. S1 – useful. Are sample weights dry or fresh?
Fig. S3 – state how many ignitions were successful for all four species. Why not present the averages for the successful ignitions – are the metrics subdued for the species with fewer successful ignitions?
Discussion S1 – Water content is potentially important for species differences, but not discussed here. If it was not important, this should be stated.
Fig. S5 – I am puzzled that flame duration aligns so closely with temperature and flame height here – in principle (and arithmetically) you would expect a negative relationship if the same mass was burnt
